# Localized Radiotherapy for Classic Kaposi’s Sarcoma: An Analysis of Lesion Characteristics and Treatment Response

**DOI:** 10.3390/cancers16183194

**Published:** 2024-09-19

**Authors:** Junhee Park, Jeong Eun Lee

**Affiliations:** Department of Radiation Oncology, School of Medicine, Kyungpook National University, Daegu 41944, Republic of Korea; jhp1247@knuh.kr

**Keywords:** Kaposi’s sarcoma, Kaposi sarcoma, radiotherapy, RT

## Abstract

**Simple Summary:**

This study aimed to evaluate the efficacy of radiotherapy for skin lesions in classic Kaposi’s sarcoma. A retrospective analysis was performed. Response after radiotherapy was defined as follows: Complete response indicated no clinically detectable skin lesions and no symptoms. Partial response was defined as a reduction in lesion height by more than half or a lighter lesion color compared to before treatment. In-field recurrence was defined as the appearance of new lesions within a previously irradiated field. The overall response rate was 100%. The efficacy of radiotherapy was evident, even in cases involving disseminated lesions. Further research on the optimal dose and fractionation is deemed necessary.

**Abstract:**

Objectives: Classic Kaposi’s sarcoma (CKS) is a rare malignancy with diverse clinical presentations, lacking a standard treatment. While localized therapies are commonly used for symptomatic lesions, radiotherapy (RT) has demonstrated effectiveness. This study aims to evaluate the efficacy of RT for treating skin lesions in CKS. Methods: A retrospective analysis was conducted on patients with KS treated between April 2012 and January 2024. In total, 69 lesions in 16 patients were included. Treatment response was defined as follows: complete response (CR) indicated the absence of clinically detectable skin lesions and symptoms; partial response (PR) was a reduction in lesion height by more than half or a lighter lesion color compared to before treatment. In-field recurrence was the appearance of new lesions within a previously irradiated field. Logistic regression analysis was used to investigate factors influencing response and in-field recurrence. Results: The median follow-up period was 52 months (range, 3–138 months). The overall response rate was 100%, with 92.8% of the patients achieving CR and 7.2% receiving PR. PR was observed in three patients with five lesions, all of which remained stable. In-field recurrence occurred in two patients with initially advanced disease, and all recurrent lesions responded to RT. No variables were significantly associated with response or in-field recurrence. Conclusions: RT for CKS showed a 100% response rate, with complete symptom relief in all cases. The effectiveness of RT was evident, even in cases involving disseminated lesions. Further research is needed to determine the optimal RT dose and fractionation.

## 1. Introduction

Kaposi’s sarcoma (KS) is a vascular endothelial malignancy, which was first described in 1872 [1]. It typically presents with multiple vascular nodules on the skin but can also involve lymph nodes (LNs), mucosal surfaces, and visceral organs. The cutaneous lesions, which are the most frequent manifestation of KS, exhibit a variety of colors and characteristics that vary depending on subtype and stage. Clinically, these lesions can appear as small papules, nodules, patches, or large plaques, often accompanied by symptoms such as pain, bleeding, and pruritus [2,3,4,5].

KS emerged as a significant concern during the human immunodeficiency virus/acquired immunodeficiency syndrome (HIV/AIDS) epidemic of the 1980s, which predominantly affected immunosuppressed individuals. The discovery of human herpesvirus 8 (HHV-8), also known as Kaposi’s sarcoma-associated herpesvirus (KSHV), in 1994 was a pivotal milestone in understanding the etiology of KS. HHV-8, a member of the Herpesviridae family, is now recognized as the primary causative agent of KS, especially in those with compromised immune systems [6,7,8].

In the management of KS, asymptomatic lesions that are few in number or occur in non-HIV-positive patients may be closely monitored. However, symptomatic lesions typically require intervention through either local or systemic therapies, depending on their location and extent. Various treatment options, including radiotherapy (RT), surgery, topical agents, cryotherapy, intralesional chemotherapy, and electrochemotherapy, are commonly used to manage KS [9,10,11,12,13,14]. Although there is no universally established treatment protocol owing to the variations in lesion type, location, and size, RT has been proven to be highly effective for the local management of KS [15,16,17,18].

This study aims to evaluate the effectiveness and safety of RT in treating HHV-8-positive classic KS (CKS) at our institution and to assess the feasibility of reducing the RT dose.

## 2. Materials and Methods

A retrospective analysis was conducted on patients with KS who were treated at Kyungpook National University Hospital between April 2012 and January 2024. The study protocol was approved by the Institutional Review Board (KNUH 2024-06-021).

All patients underwent a dermatologic evaluation followed by a biopsy to confirm the pathological diagnosis of KS. Imaging studies were then performed for staging purposes, with either CT or PET CT used based on the extent of the lesions. In addition, serum HIV antibody testing and HHV-8 immunohistochemistry (IHC) or polymerase chain reaction (PCR) were performed for all patients. Only patients who tested positive for HHV-8 on IHC or PCR were included in the study, whereas those who were HIV seropositive or had a history of receiving other systemic therapies for KS were excluded.

Data collected from medical records included sex, age at diagnosis, lesion characteristics, symptoms, RT methods, RT dose, RT fractionation, treatment responses, side effects, and the date of the last follow-up.

Response after RT was defined as follows: complete response (CR) indicated the absence of clinically detectable skin lesions and symptoms, and partial response (PR) was defined as a reduction in lesion height by more than half or a lightening of the lesion color compared to before treatment. The overall response rate after RT for KS was calculated as the sum of the CR and PR rates. In-field recurrence was defined as the appearance of new lesions within a previously irradiated field.

The primary endpoint was the effectiveness of RT for KS. Logistic regression analysis was used to investigate the factors that influenced treatment response and the occurrence of in-field recurrence.

### Radiotherapy

At our hospital, RT was administered using electron or 6-megavoltage photon beams, with the choice of technique and energy tailored to the size and location of the lesions. For electron treatments, the RT field was determined based on lesion size, with a 1–2 cm margin added around each lesion. In cases involving extensive or circumferential plaque lesions, electron therapy alone was often inadequate. Therefore, photon treatment plans, utilizing methods such as 3D-conformal RT, were employed to ensure comprehensive coverage. Bolus materials were used to optimize skin dose coverage during irradiation.

## 3. Results

### 3.1. Patient Characteristics

Out of 26 patients diagnosed with KS, 16 were included in the study, representing a total of 69 lesions analyzed. Ten patients were excluded for the following reasons: loss of follow-up without RT (four patients), RT not administered at our institution (three patients), diagnosis of epidemic KS with AIDS treated with highly active antiretroviral therapy (one patient), surgical excision of a tonsillar lesion (one patient), and chemotherapy for disseminated stage IV disease (one patient). We excluded patients with epidemic KS because there was only one patient in our hospital, and we believed that the clinical course might differ due to the administration of highly active antiretroviral therapy. None of the included patients had a history of other cancers or treatments that could cause immunosuppression. Three patients were diagnosed with diabetes, but all were well controlled without chronic uncontrolled conditions, thereby classified as CKS. At diagnosis, the median age of the patients (12 males, 4 females) was 75 years (range, 32–86 years). In total, 8 out of the 16 patients experienced initial symptoms, primarily induration, tenderness, and itching sensation.

One patient presented with a 0.5 cm-sized mass at diagnosis, which was difficult to sample using a punch biopsy, and thus, underwent an excisional biopsy for both diagnostic and therapeutic purposes. After a period of observation without further lesions, recurrence was confirmed at the same site after 1 year, leading to the initiation of RT. The remaining 15 patients received RT without prior local treatment. The patient characteristics are detailed in Table 1.

### 3.2. Radiotherapy

The median dose was 30 Gy (range, 20–54 Gy), using fraction sizes of 1.8–5 Gy. The biologically effective dose 10 (BED10) to the tumor was calculated using an α/β ratio of 10 Gy, accounting for the varied doses and fractionation schemes used in the treatment, with a median BED10 of 39.0 Gy_10_ (range, 27.3–70.2 Gy_10_). Treatment plans were selected based on the location and depth of the lesion, with 36 lesions treated using electron beams and 33 with photon beams (Table 2).

Electron beam energy was selected to ensure the coverage of 85–90% of the lesions. Bolus material was applied in 58 (84.1%) cases to ensure adequate coverage by the radiation beam. One patient with multiple lesions on both feet was treated using an acrylic box filled with water. Lesions were predominantly located on the lower extremities (49.3%), upper extremities (46.4%), face (2.9%), and abdomen (1.5%). Detailed case information is provided in Table 3.

### 3.3. Tumor Response

The median duration of follow-up was 52 months (range, 3–138 months). All symptoms had resolved completely at the time of follow-up. The overall response rate for all treated lesions was 100%, with 92.8% (64/69) of them achieving CR and 7.2% (5/69) showing PR. The PR lesions were observed in three patients with five lesions. Specifically, two patients received RT with 30 Gy in 10 fractions to the right elbow and left hand, and the third patient received RT with 25 Gy in 10 fractions to the right forearm, right hand, and left forearm. All these PR lesions remained stable without progression until the last follow-up.

In-field recurrence was observed in two patients who were initially diagnosed with stage IV disease. Despite the presence of widespread lesions, these patients opted for localized treatment, prioritizing RT owing to symptom severity and lesion location.

One patient, diagnosed at the age of 64 years, initially presented with lesions on both the lower and upper extremities. This patient received RT at a dose of 30 Gy in 10 fractions for a plaque on the left thigh. After 9 months, multiple nodular lesions appeared on the left thigh, prompting re-irradiation with 25 Gy in 10 fractions. Following re-irradiation, the patient achieved a CR and remained disease-free at the treated site for 66 months.

The other patient, diagnosed with disseminated disease at the age of 81 years, underwent RT for symptomatic lesions on both forearms, both hands, and the face, achieving CR for the treated lesions. However, 10 months after RT, new lesions appeared on the right forearm and the dorsa of both feet, outside the RT field, and were confirmed as recurrences via punch biopsy. Despite the presence of multiple recurrent lesions, the patient declined further treatment and was lost to follow-up.

Two years later, the progression of untreated lesions led to the development of painful and bleeding symptoms, prompting the patient to seek medical care again. Owing to financial constraints, the patient declined further diagnostic tests and systemic treatment, opting instead for localized RT targeting symptomatic areas. These lesions recurred within the previous RT field after 52 months (Figure 1). RT with 25 Gy in 10 fractions was administered to both forearms, resulting in a PR. The re-irradiated lesions remained in the PR state for 7 months.

Two of the treated patients died from causes unrelated to their KS. One patient died from pneumonia following total hip replacement surgery, and the other patient died from an intracranial hemorrhage resulting from trauma. Both patients had multiple lesions on both the upper and lower extremities but showed no evidence of disease after RT, having achieved a CR state.

Variables that might affect treatment response, such as lesion location, recurrent lesions, number of lesions, RT method, and BED10, were evaluated. None of these variables were found to be associated with treatment response or recurrence.

### 3.4. Toxicity

The adverse effects observed in 69 RT-treated lesions were documented according to the Common Terminology Criteria for Adverse Events version 5.0 and are detailed in Table 4. The most frequently observed side effect following RT was dry desquamation of the skin (63.8%), followed by skin induration (36.2%) and hyperpigmentation (31.9%). Grade 2 side effects were observed in four (5.8%) cases of lymphedema and two (2.9%) cases of dry desquamation, all of which showed improvement during follow-up. No Grade 3 or higher side effects were reported.

## 4. Discussion

KS is categorized into four epidemiological forms based on ethnicity, region, HIV infection, and immunological status. CKS is associated with ethnicity and is primarily found in Jewish men of Ashkenazi descent or individuals living near the Mediterranean region. It predominantly affects the lower extremities of elderly men, and a previous HHV-8 infection is a necessary condition [2,10]. Endemic KS, which is more aggressive than CKS, frequently involves LNs and is prevalent in parts of Central and Eastern Africa [9]. Epidemic KS, also known as AIDS-associated KS (AKS), was first reported in the 1980s and occurs in HIV-positive patients [19]. The widespread adoption of highly active antiretroviral therapy in the early to mid-1990s led to a significant annual decline in both the incidence and mortality rates of KS [20,21,22,23]. Iatrogenic KS occurs in individuals with immunosuppression, such as those undergoing immunosuppressive treatment or organ transplantation. The epidemiological investigation of AKS led to the identification of a previously unknown herpes virus, which was subsequently demonstrated to be associated with all forms of KS. HHV-8, also known as KSHV, was first discovered in 1994 by Chang et al. using molecular techniques during the examination of KS lesions in patients with AIDS [24].

KS requires a multidisciplinary approach. Due to its skin manifestations, dermatologists are crucial for accurate evaluation and monitoring. However, KS can affect not only the skin and mucous membranes but also internal organs and lymph nodes, so a multidisciplinary approach involving specialists from various fields, including radiation oncologists and chemotherapy experts, is necessary. A study by Benajiba et al. identified several risk factors for initiating systemic treatment in classical and endemic KS, including a time interval of more than 1 year between symptom onset and diagnosis, endemic KS, a total number of lesions exceeding 10, localization to internal organs and the head and neck, and the presence of edema [25]. These findings highlight the need for a tailored approach based on individual patient characteristics. The treatment of KS often involves RT, chemotherapy, and, in cases such as AKS, a combination of antiretroviral therapy, each requiring specific expertise. A multidisciplinary team is crucial for developing an integrated treatment plan, enhancing diagnostic accuracy, and more effectively managing treatment-related side effects and complications.

Treatment for KS is individualized depending on the type and extent of lesions. A classification system has been developed to stage CKS, taking into account lesion distribution and disease progression [5]. The four clinical forms of KS are categorized as follows: Stage I is localized nodular KS; Stage II is localized, invasive, and aggressive KS; Stage III is disseminated mucocutaneous KS; and Stage IV is Stage III with visceral involvement.

Localized symptomatic lesions often respond well to local treatments such as RT, excision, topical agents, cryotherapy, intralesional chemotherapy, and electrochemotherapy [11,12,13]. RT is widely recognized as the primary treatment for localized KS, with reported overall response rates of up to 100% and CR rates ranging from 30% to 90% [3,26,27]. In addition to conventional RT, brachytherapy is a viable treatment option for localized skin KS. Kasper et al. reported that applying 24–35 Gy in four to six fractions using high-dose-rate brachytherapy for 16 patients resulted in a 100% CR rate [28]. Similarly, Ruiz et al. achieved a 100% CR rate with 5 Gy in five fractions of brachytherapy applied to five lesions, also noting favorable cosmetic outcomes [29]. In our study, localized symptomatic lesions treated with RT showed excellent efficacy, achieving a 100% overall response rate and complete symptom resolution. The majority of lesions were located in the extremities, with 75.0% of cases presenting as multifocal.

In our logistic regression analysis to identify prognostic factors, it was challenging to find significant associations between the characteristics of KS and the outcomes of RT. Neither the number of tumor lesions at the time of RT nor the BED had a significant impact on tumor response or in-field recurrence. This lack of significant effect might be attributed to the high sensitivity of KS to RT, which could diminish the influence of these factors. Additionally, the RT dose used in this study may have been sufficient to achieve the maximum therapeutic effect, minimizing the impact of BED or lesion number on treatment response.

Several studies have compared different radiation doses in the treatment of KS. A study conducted by Wulf et al. aimed to evaluate the efficacy of low-dose RT for the treatment of AKS from 1983 to 1990, before the introduction of highly active antiretroviral therapy [30]. This study initially treated 74 cutaneous KS lesions with RT of 2 Gy per fraction three times, achieving a 70% success rate. The efficacy was significantly improved when the dose was increased to 4 Gy per fraction three times. In a larger cohort of 2066 skin KS lesions, the 4 Gy regimen achieved a significantly higher success rate of 93%. In addition, 165 mucosal KS lesions treated with 4 Gy per fraction three times also achieved a high success rate of 91%. These results suggest that low-dose RT, especially the 4 Gy regimen, is very effective in the management of cutaneous and mucosal KS lesions. A prospective study by Yildiz et al. demonstrated comparable overall response rates between 8-Gy and 6-Gy treatments, although a statistically significant difference in CR rates was observed at 12 months. Patients receiving an 8-Gy dose showed a CR rate of 93%, whereas those receiving a 6-Gy dose showed a CR rate of 60% [31]. Another study comparing 16 Gy delivered in four fractions over 4 days with a single fraction of 8 Gy found no statistically significant differences in overall response or skin pigmentation for AKS [32]. However, other research has indicated that higher doses are associated with improved response rates and extended tumor control. A prospective randomized study including 71 AKS lesions compared 8 Gy in a single fraction, 20 Gy in 10 fractions, and 40 Gy in 20 fractions and reported that fractionated RT to higher doses improved CR rates, lowered the incidence of residual pigmentation and caused a longer median time to treatment failure [33]. Despite these findings, it has been suggested that higher doses should be tailored to the individual needs of each patient. In addition, a retrospective study reported 5-year CR rates of 91.6% with doses greater than 20 Gy and 89.6% with doses of 8 Gy [16]. In a prospective randomized trial, hypofractionated RT with 20 Gy in 5 fractions was compared to conventional RT with 24 Gy in 12 fractions, showing similar response rates, local control, and skin toxicity between the two groups [34].

Given the variety of dose regimens used in treatment, direct comparisons are challenging. The simple application of BED based on the linear–quadratic model may not fully capture the effects, especially with high doses per fraction. However, evaluating outcomes using BED or equivalent doses in 2-Gy fractions (EQD2) could provide valuable insights. Oysul et al. reported in their retrospective analysis that local RT with an EQD2 of 20 Gy or higher significantly improved CR rates in CKS, compared to RT with an EQD2 of less than 20 Gy [35]. Furthermore, RT has been shown to improve symptoms effectively. Administering total doses ranging from 8 to 12 Gy in a single fraction, or 24 to 30 Gy over 2–3 weeks in two fractions, consistently provided substantial symptomatic relief in 80–100% of cases, particularly for symptoms such as mucositis associated with epidemic KS [36].

In summary, various dose regimens in RT for KS have yielded effective treatment outcomes, with no significant differences in side effects observed among different regimens.

Despite the limitations of this study owing to its retrospective nature, small sample size, and being conducted at a single institution, it is nonetheless valuable in analyzing the RT response and required RT dose of CKS in the Korean population.

## 5. Conclusions

In this study, we evaluated the outcomes of RT administered to 69 lesions in 16 patients with CKS. The overall response rate was 100%, with complete symptom relief observed in every case. The efficacy of RT in treating KS was evident, even in cases involving disseminated lesions. However, CKS is prone to local recurrence, as confirmed by our data. Owing to the nature of CKS, there is a concern regarding potential overlap in RT fields during subsequent treatments. Given these characteristics and the favorable response to RT, lower-dose RT can be considered. Further research is needed to determine the optimal RT dose and fractionation.

## Figures and Tables

**Figure 1 cancers-16-03194-f001:**
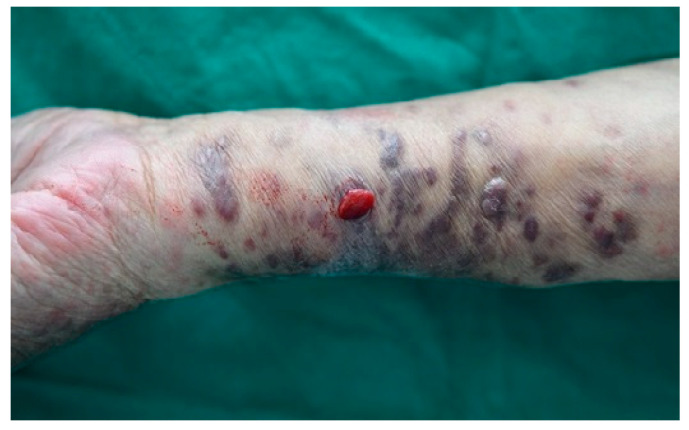
In-field recurrence in a patient who had initially achieved complete response after receiving 30 Gy of radiotherapy in 10 fractions.

**Table 1 cancers-16-03194-t001:** Patient characteristics (*n* = 16).

Characteristics		Numbers (%)
Age at diagnosis (year)	Median (range)	75 (32–86)
Sex	Male	12 (75.0)
	Female	4 (25.0)
Prior treatment	Excision	1 (6.2)
	No	15 (93.8)
Stage	1	4 (25.0)
	2	5 (31.3)
	3	3 (18.7)
	4	4 (25.0)
Multiplicity of skin lesions	Single	4 (25.0)
	Multiple	12 (75.0)
No. of RT sessions	1	10 (62.5)
	2	3 (18.8)
	3	1 (6.2)
	4 and more	2 (12.5)

“No. of RT sessions” indicates the number of radiotherapy sessions administered to each individual patient. For the 16 patients treated, those who experienced recurrence during follow-up underwent multiple rounds of radiotherapy. Abbreviations: RT, radiotherapy.

**Table 2 cancers-16-03194-t002:** Lesion characteristics (lesions = 69).

Characteristics		Numbers (%)
Location	Upper extremities	32 (46.4)
	Lower extremities	34 (49.3)
	Face	2 (2.9)
	Abdomen	1 (1.5)
RT method	Photon	33 (47.8)
	Electron	36 (52.2)
BED10 (Gy_10_)	<39.0	31 (44.9)
	≥39.0	38 (55.1)
Clinical morphology	Papule	18 (26.1)
	Nodule	20 (29.0)
	Plaque	31 (44.9)

Abbreviations: RT, radiotherapy; BED10, biologically effective dose for tumor cell kill with α/β ratio of 10 Gy.

**Table 3 cancers-16-03194-t003:** Details of cases.

Site	Dose (Gy) Median (Range)	Fractions Median (Range)	BED10 (Gy_10_) Median (Range)	EQD2 (Gy) Median (Range)	Type of Response
Lt lower extremity ^1^	30 (25–30)	10 (5–15)	39.0 (28.0–39.0)	32.5 (23.3–32.5)	CR (15 cases)
Rt lower extremity ^1^	30 (20–50.4)	10 (5–28)	39.0 (28.0–59.5)	32.5 (23.3–49.6)	CR (20 cases)
Lt upper extremity ^2^	30 (20–54)	10 (5–18)	39.0 (28.0–70.2)	32.5 (23.3–58.5)	CR (14 cases) PR (2 cases)
Rt upper extremity ^2^	30 (21–30)	10 (7–15)	39.0 (27.3–39.0)	32.5 (26.0–32.5)	CR (12 cases) PR (3 cases)
Trunk	30	10	39.0	32.5	CR
Face ^3^	30	15	36.0	30.0	CR (2 cases)

^1^ Lower extremities include toe, sole, foot, shin, calf, and thigh. ^2^ Upper extremities include finger, hand, forearm, and upper arm. ^3^ Facial lesions include left nose root and right zygoma. Both lesions were irradiated with the same dose fractionation schedule. Abbreviations: BED10, biologically effective dose for killing tumor cell kills with an α/β ratio of 10 Gy; EQD2, equivalent dose in 2-Gy fractions; Lt, left; Rt, right; CR, complete response; PR, partial response.

**Table 4 cancers-16-03194-t004:** Adverse effects (*n* = 69).

	Grade 0	Grade 1	Grade 2
Dry desquamation	23 (33.3)	44 (63.8)	2 (2.9)
Pain of skin	57 (82.6)	12 (17.4)	0 (0.0)
Hyperpigmentation	47 (68.1)	22 (31.9)	0 (0.0)
Skin induration	44 (63.7)	25 (36.2)	0 (0.0)
Lymphedema	50 (72.5)	15 (21.7)	4 (5.8)
Bullous dermatitis	68 (98.5)	1 (1.5)	0 (0.0)
Nail loss	67 (97.1)	2 (2.9)	0 (0.0)

## Data Availability

The data underlying this article will be shared upon reasonable request to the corresponding author.

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
