# Peer review of "Localized Radiotherapy for Classic Kaposi’s Sarcoma: An Analysis of Lesion Characteristics and Treatment Response"

_cancers, 2024, doi:10.3390/cancers16183194_

Round 1
Reviewer 1 Report
Comments and Suggestions for Authors
The authors present a case series describing the outcomes of cutaneous KS (CKS) upon localized radiotherapy. A cases series is reasonable for this disease given its rarity and absence of high quality clinical trials to evaluate the various treatment modalities and approaches. The findings indicate dramatic response of CKS to radiotherapy. As the authors appropriately indicate, these findings should be used to evaluate lower doses of radiotherapy and confirm appropriate fractionation schedules. It is not clear why the authors excluded patients with HIV infection but if trialed in such higher incidence populations, radiotherapy can provide a toxicity-avoiding alternative to chemotherapy or in combination. Please explain the rationale for the exclusion of HIV-infected patients.
Author Response
Dear Editors and Reviewers,
We thank the editors and reviewers for their careful review of our manuscript with valuable comments and suggestions. A substantial revision of the manuscript has been conducted to take all of them into account. We have done our best to address all the points raised. We believe the manuscript has been significantly improved and hope that it is suitable for consideration for the publication. Each comment is followed by the corresponding answer highlighted in red. In addition, we made corrections and clarifications as the editor and reviewers suggested, all of them are highlighted with ‘track changes’ feature in Microsoft Word within the revised manuscript.
Comments and Suggestions for Authors
The authors present a case series describing the outcomes of cutaneous KS (CKS) upon localized radiotherapy. A cases series is reasonable for this disease given its rarity and absence of high quality clinical trials to evaluate the various treatment modalities and approaches. The findings indicate dramatic response of CKS to radiotherapy. As the authors appropriately indicate, these findings should be used to evaluate lower doses of radiotherapy and confirm appropriate fractionation schedules. It is not clear why the authors excluded patients with HIV infection but if trialed in such higher incidence populations, radiotherapy can provide a toxicity-avoiding alternative to chemotherapy or in combination. Please explain the rationale for the exclusion of HIV-infected patients.
[Response]
Thank you for your comment. The age-standardized incidence of Kaposi sarcoma (KS) in skin cancer in Korea from 2006 to 2018 was 0.28 [1], and only 20 cases of KS were found in HIV-positive patients from 2006 to 2018 [2]. In addition, we conducted the study on classic KS only because we had only one patient from our hospital, and it is important to perform HIV treatment such as highly active antiretroviral therapy in the case of AIDS-related KS, and it is thought that the pattern may be different because of this.
Revised sentence:
(Materials and Methods, page 3)
We excluded patients with epidemic KS because there was only one patient in our hospital, and we believed that the clinical course might differ due to the administration of highly active antiretroviral therapy.
- Park, K.; Bae, J.M.; Chung, K.Y.; Yun, S.J.; Seo, S.H.; Ahn, H.H.; Lee, D.Y.; Kim, H.; Sohn, U.; Park, B.C. Incidence and Prevalence of Skin Cancers in South Korea from 2008 to 2016: A Nation-Wide Population Based Study. Ann Dermatol 2022, 34, 105-109, doi:10.5021/ad.2022.34.2.105.
- Park, B.; Ahn, K.H.; Choi, Y.; Kim, J.H.; Seong, H.; Kim, Y.J.; Choi, J.Y.; Song, J.Y.; Lee, E.; Jun, Y.H.; et al. Cancer Incidence Among Adults With HIV in a Population-Based Cohort in Korea. JAMA Netw Open 2022, 5, e2224897, doi:10.1001/jamanetworkopen.2022.24897.
Reviewer 2 Report
Comments and Suggestions for Authors
The authors provided a paper about “Localized radiotherapy for classic Kaposi’s sarcoma: An analysis of lesion characteristics and treatment response”.
The topic is interesting both from a clinical and a research perspective.
Over it is a good manuscript well written and addresses several relevant points of this topic.
I have a few points that I would like the authors to address in order to improve the scientific soundness of this paper:
1) Please discuss the possible strategies for cases of KS with nodal or organ involvement
2) Please add data about the imaging used to stage the patients in the present series (CT? MRI? PET-CT? Only clinical evaluation)
3) Since most KS lesions are superficial also brachytherapy is used in several centers (please refer to PMID: 37425199 for technical details)
4) Please discuss more in detail the radiosensitivity of KS and the relative low doses needed to treat it
5) Please add a paragraph about the need for multidisciplinary management of these patients tighter with dermatologists
6) Please provide reasons for the different fractionations used in the present series both in terms of overall dose and fractionation
7) In table 1 lease explain better is meant by “No. of RT sessions”
8) In table 3 there is no need to present all the single cases, group them according to anatomical locations and provide mean values with ranges of the other values in the nearby columns
Author Response
Dear Editors and Reviewers,
We thank the editors and reviewers for their careful review of our manuscript with valuable comments and suggestions. A substantial revision of the manuscript has been conducted to take all of them into account. We have done our best to address all the points raised. We believe the manuscript has been significantly improved and hope that it is suitable for consideration for the publication. Each comment is followed by the corresponding answer highlighted in red. In addition, we made corrections and clarifications as the editor and reviewers suggested, all of them are highlighted with ‘track changes’ feature in Microsoft Word within the revised manuscript.
Comments and Suggestions for Authors
The authors provided a paper about “Localized radiotherapy for classic Kaposi’s sarcoma: An analysis of lesion characteristics and treatment response”.
The topic is interesting both from a clinical and a research perspective.
Over it is a good manuscript well written and addresses several relevant points of this topic.
I have a few points that I would like the authors to address in order to improve the scientific soundness of this paper:
[Response] Thank you for your good review of the paper. I will comment on the review you mentioned.
1) Please discuss the possible strategies for cases of KS with nodal or organ involvement.
[Response]
Thank you for your comment. This study analyzed the effect of radiotherapy (RT) on cutaneous classic Kaposi’s sarcoma (CKS), and did not cover cases with invasion to other organs or lymph nodes. Among the KS patients registered in our hospital from 2006 to 2018, only 1 patient out of 26 had invasion to other organs or lymph nodes, and that patient was treated with surgery, so we excluded that patient from this analysis. In addition, if there are patients with such invasion, RT can be considered depending on the patient’s overall condition and the extent and location of the lesion, and chemotherapy can also be used. However, this is not related to the scope of this study, so it would be better to conduct a new study targeting these patients. In addition to comment 5. We added in the discussion that multidisciplinary treatment is necessary and that various treatments may be needed depending on the extent of the lesion.
Revised sentence:
(Discussion, page 6-7)
KS requires a multidisciplinary approach. Due to its skin manifestations, dermatologists are crucial for accurate evaluation and monitoring. However, KS can affect not only the skin and mucous membranes but also internal organs and lymph nodes, a multidisciplinary approach involving specialists from various fields, including radiation oncologists and chemotherapy experts, is necessary. A study by Benajiba et al. identified several risk factors for initiating systemic treatment in classical and endemic KS, including a time interval of more than 1 year between symptom onset and diagnosis, endemic KS, a total number of lesions exceeding 10, localization to internal organs and the head and neck, and the presence of edema. These findings highlight the need for a tailored approach based on individual patient characteristics. Treatment of KS often involves radio-therapy, chemotherapy, and, in cases such as AKS, a combination of antiretroviral therapy, each requiring specific expertise. A multidisciplinary team is crucial for developing an integrated treatment plan, enhancing diagnostic accuracy, and more effectively managing treatment related side effects and complications.
2) Please add data about the imaging used to stage the patients in the present series (CT? MRI? PET-CT? Only clinical evaluation)
[Response]
Thank you for your comment. We added details to materials and methods as you pointed out. Imaging studies were then performed for staging purposes, with either CT or PET CT used based on the extent of the lesions.
Revised sentence:
(Materials and Methods, page 2)
Imaging studies were then performed for staging purposes, with either CT or PET CT used based on the extent of the lesions.
3) Since most KS lesions are superficial also brachytherapy is used in several centers (please refer to PMID: 37425199 for technical details)
[Response]
Thank you for your comment. I understand that brachytherapy is a valuable treatment for skin cancer. In Korea, superficial brachytherapy is not commonly used to treat skin cancers, and no patient used superficial brachytherapy in this cutaneous CKS patient. I will include information about brachytherapy and its benefits among the treatment options in the discussion section.
Revised sentence:
(Discussion, page 7)
In addition to conventional RT, brachytherapy is a viable treatment option for localized skin KS. Kasper et al. reported that applying 24–35 Gy in 4–6 fractions using high-dose-rate brachytherapy for 16 patients resulted in a 100% CR rate. Similarly, Ruiz et al. achieved a 100% CR rate with 5 Gy in 5 fractions of brachytherapy applied to 5 lesions, also noting favorable cosmetic outcomes.
4) Please discuss more in detail the radiosensitivity of KS and the relative low doses needed to treat it.
[Response]
Thank you for your advice. We included additional studies in the discussion to provide a more comprehensive analysis.
Revised sentence:
(Discussion, page 7-8)
A study conducted by Wulf et al. aimed to evaluate the efficacy of low dose RT for the treatment of AKS from 1983 to 1990, before the introduction of highly active antiretroviral therapy. This study initially treated 74 cutaneous KS lesions with RT of 2 Gy per fraction with three times, achieving a 70% success rate. The efficacy was significantly improved when the dose was increased to 4 Gy per fraction with three times. In a larger cohort of 2,066 skin KS lesions, the 4 Gy regimen achieved a significantly higher success rate of 93%. In addition, 165 mucosal KS lesions treated with 4 Gy per fraction with three times also achieved a high success rate of 91%. These results suggest that low dose RT, especially the 4 Gy regimen, is very effective in the management of cutaneous and mucosal KS lesions.
Another study comparing 16 Gy delivered in 4 fractions over 4 days with a single fraction of 8 Gy found no statistically significant differences in overall response or skin pigmentation for AKS. However, other research has indicated that higher doses are associated with improved response rates and extended tumor control.
A prospective randomized study including 71 AKS lesions compared 8 Gy in a single fraction, 20 Gy in 10 fractions, and 40 Gy in 20 fractions and reported that fractionated RT to higher doses im-proved CR rates, lower incidence of residual pigmentation and longer median time to treatment failure. Despite these findings, it has been suggested that higher doses should be tailored to the individual needs of each patient.
5) Please add a paragraph about the need for multidisciplinary management of these patients tighter with dermatologists
[Response]
Thank you for your comment. We added details to discussion as you pointed out.
Revised sentence:
(Discussion, page 6-7)
KS requires a multidisciplinary approach. Due to its skin manifestations, dermatologists are crucial for accurate evaluation and monitoring. However, because KS can affect not only the skin and mucous membranes but also internal organs and lymph nodes, a multidisciplinary approach involving specialists from various fields, including radiation oncologists and chemotherapy experts, is essential. A study by Benajiba et al. identified several risk factors for initiating systemic treatment in classical and endemic KS, including a delay of more than 1 year between symptom onset and diagnosis, endemic KS, a total number of lesions exceeding 10, localization to internal organs and the head and neck, and the presence of edema. These findings highlight the need for a tailored approach based on individual patient characteristics. Treatment of KS often involves radiotherapy, chemotherapy, and, in cases such as AKS, a combination of antiretroviral therapy, each requiring specific expertise. A multidisciplinary team is crucial for developing an integrated treatment plan, enhancing diagnostic accuracy, and more effectively managing treatment related side effects and complications.
6) Please provide reasons for the different fractionations used in the present series both in terms of overall dose and fractionation
[Response]
Thank you for your comment. Various dose regimen was used. Several doctors performed RT for various lesions over a long period of time from 2006 to 2018, so it is thought that there are various fractionations of treatment. If the skin lesions were extensive and the lesion thickness was large, high-dose RT was performed. In cases of relapse or extremity lesions such as fingers and toes, low dose fractionations were used considering that side effects may occur when the dose per fraction is high. Due to the limitations of retrospective nature, a study was planned to analyze the good response shown even with low dose RT.
7) In table 1 lease explain better is meant by “No. of RT sessions”
[Response]
Thank you for your advice. "No. of RT sessions" refers to the number of radiotherapy sessions administered to each individual patient. After treating 16 patients, some of whom experienced recurrence during follow-up, multiple rounds of treatment were required for these patients. This detail will be clarified in the explanatory notes below the table.
Revised table:
(Results, page 3)
1"No. of RT sessions" indicates the number of radiotherapy sessions administered to each individual patient. For the 16 patients treated, those who experienced recurrence during follow-up underwent multiple rounds of radiotherapy.
8) In table 3 there is no need to present all the single cases, group them according to anatomical locations and provide mean values with ranges of the other values in the nearby columns
[Response]
Thank you for comment. I have updated the table as you suggested.
Revised table:
(Results, page 4)
Table 3. Details of cases.
|
Site |
Dose (Gy) |
Fractions |
BED10 (Gy10) median (range) |
EQD2 (Gy) median (range) |
Type of |
|
Lt lower extremity1 |
30 (25–30) |
10 (5–15) |
39.0 (28.0–39.0) |
32.5 (23.3–32.5) |
CR (15 cases) |
|
Rt lower extremity1 |
30 (20–50.4) |
10 (5–28) |
39.0 (28.0–59.5) |
32.5 (23.3–49.6) |
CR (20 cases) |
|
Lt upper extremity2 |
30 (20–54) |
10 (5–18) |
39.0 (28.0–70.2) |
32.5 (23.3–58.5) |
CR (14 cases) PR (2 cases) |
|
Rt upper extremity2 |
30 (21–30) |
10 (7–15) |
39.0 (27.3–39.0) |
32.5 (26.0–32.5) |
CR (12 cases) PR (3 cases) |
|
Trunk |
30 |
10 |
39.0 |
32.5 |
CR |
|
Face3 |
30 |
15 |
36.0 |
30.0 |
CR (2 cases) |
1Lower extremities include toe, sole, foot, shin, calf and thigh.
2Upper extremities include finger, hand, forearm and upper arm.
3Facial lesions include left nose root and right zygoma. Both lesions were irradiated with the same dose fractionation schedule.
BED10, biologically effective dose for killing tumor cell kills with an α/β ratio of 10 Gy; EQD2, equivalent dose in 2-Gy fractions; Lt, Left; Rt, Right; CR, complete response; PR, partial response.
Reviewer 3 Report
Comments and Suggestions for Authors
This manuscript is a retrospective analysis of the efficacy of radiotherapy for classic Kaposi sarcoma in a single cohort of 16 individuals from the Korean population. In addition, the authors provide a comprehensive review of similar studies on the application of radiotherapy and their results.
The study is interesting, however, there are some concerns to discuss:
1. As far as can be deduced from the article, classic Kaposi sarcoma is predominantly typical for individuals of Mediterranean or Ashkenazi Jewish origin. If the cohort of this study is predominantly Korean, the manuscript could be supplemented with incidence statistics specific to your region to complete the study. Perhaps material from a recent publication would help to supplement and expand this issue [DOI: 10.1016/S2214-109X(23)00349-2].
2. The manuscript cites references mostly to publications from 10 to 15 years ago. Partly, this is due to the fact that the author provides a results summary of specific studies on the use of radiotherapy for the treatment of Kaposi sarcoma, the list of which isn't large on its own. Even though the analysis of general information in this article includes references to recent publications, it would have been appropriate to supplement the list of sources with more recent data from systematic reviews. For example: [DOI:0.11604/pamj-cm.2024.15.13.43128], [DOI:10.3390/cancers14081915]
3. While the manuscript analyzed data for each individual neoplasia (n=69), is the sample size among patients (n=16) sufficient to validate any results on the efficacy of the proposed radiotherapy application technique?
4. Even though the authors acknowledge a number of limitations of the study, the study itself emphasizes that «In our logistic regression analysis to identify prognostic factors, it was challenging to find significant associations between the characteristics of KS and the outcomes of RT», it should be asked what is the importance of the published data and its usefulness in improving Kaposi sarcoma therapy?
Author Response
Dear Editors and Reviewers,
We thank the editors and reviewers for their careful review of our manuscript with valuable comments and suggestions. A substantial revision of the manuscript has been conducted to take all of them into account. We have done our best to address all the points raised. We believe the manuscript has been significantly improved and hope that it is suitable for consideration for the publication. Each comment is followed by the corresponding answer highlighted in red. In addition, we made corrections and clarifications as the editor and reviewers suggested, all of them are highlighted with ‘track changes’ feature in Microsoft Word within the revised manuscript.
This manuscript is a retrospective analysis of the efficacy of radiotherapy for classic Kaposi sarcoma in a single cohort of 16 individuals from the Korean population. In addition, the authors provide a comprehensive review of similar studies on the application of radiotherapy and their results.
The study is interesting, however, there are some concerns to discuss:
[Response] Thank you for your positive review of our paper.
- As far as can be deduced from the article, classic Kaposi sarcoma is predominantly typical for individuals of Mediterranean or Ashkenazi Jewish origin. If the cohort of this study is predominantly Korean, the manuscript could be supplemented with incidence statistics specific to your region to complete the study. Perhaps material from a recent publication would help to supplement and expand this issue [DOI: 10.1016/S2214-109X(23)00349-2].
[Response]
Thank you for your comment. The age-standardized incidence of Kaposi sarcoma (KS) in skin cancer in Korea from 2006 to 2018 was 0.28 [1], and only 20 cases of KS were found in HIV-positive patients from 2006 to 2018 [2]. In addition, we conducted the study on classic KS only because we had only one patient from our hospital, and it is important to perform HIV treatment such as highly active antiretroviral therapy in the case of AIDS-related KS (or epidemic KS), and it is thought that the pattern may be different because of this.
Revised sentence:
(Materials and Methods, page 3)
We excluded patients with epidemic KS because there was only one patient in our hospital, and we believed that the clinical course might differ due to the administration of highly active antiretroviral therapy.
- The manuscript cites references mostly to publications from 10 to 15 years ago. Partly, this is due to the fact that the author provides a results summary of specific studies on the use of radiotherapy for the treatment of Kaposi sarcoma, the list of which isn't large on its own. Even though the analysis of general information in this article includes references to recent publications, it would have been appropriate to supplement the list of sources with more recent data from systematic reviews. For example: [DOI:0.11604/pamj-cm.2024.15.13.43128], [DOI:10.3390/cancers14081915]
[Response]
Thank you for your suggestion. We included additional studies in the discussion to provide a more comprehensive analysis. We changed the order of the contents in the text
Revised sentence:
(Discussion, page 7-8)
A study conducted by Wulf et al. aimed to evaluate the efficacy of low dose RT for the treatment of AKS from 1983 to 1990, before the introduction of highly active antiretroviral therapy. This study initially treated 74 cutaneous KS lesions with RT of 2 Gy per fraction with three times, achieving a 70% success rate. The efficacy was significantly improved when the dose was increased to 4 Gy per fraction with three times. In a larger cohort of 2,066 skin KS lesions, the 4 Gy regimen achieved a significantly higher success rate of 93%. In addition, 165 mucosal KS lesions treated with 4 Gy per fraction with three times also achieved a high success rate of 91%. These results suggest that low dose RT, especially the 4 Gy regimen, is very effective in the management of cutaneous and mucosal KS lesions.
Another study comparing 16 Gy delivered in 4 fractions over 4 days with a single fraction of 8 Gy found no statistically significant differences in overall response or skin pigmentation for AKS. However, other research has indicated that higher doses are associated with improved response rates and extended tumor control.
A prospective randomized study including 71 AKS lesions compared 8 Gy in a single fraction, 20 Gy in 10 fractions, and 40 Gy in 20 fractions and reported that fractionated RT to higher doses im-proved CR rates, lower incidence of residual pigmentation and longer median time to treatment failure. Despite these findings, it has been suggested that higher doses should be tailored to the individual needs of each patient.
- While the manuscript analyzed data for each individual neoplasia (n=69), is the sample size among patients (n=16) sufficient to validate any results on the efficacy of the proposed radiotherapy application technique?
[Response]
Thank you for your comment. Although the number of patients was limited, the study highlights the role of RT in local control, as evidenced by the management of 69 individual lesions. The fact that a patient, despite undergoing multiple recurrences, achieved effective control of each lesion through RT and experienced symptomatic relief is noteworthy.
Further research into low dose regimens is warranted. However, the effectiveness of the low dose RT for recurrence at the same site or overlapping the previous treatment field underscores the potential benefits of RT in controlling these conditions.
- Even though the authors acknowledge a number of limitations of the study, the study itself emphasizes that «In our logistic regression analysis to identify prognostic factors, it was challenging to find significant associations between the characteristics of KS and the outcomes of RT», it should be asked what is the importance of the published data and its usefulness in improving Kaposi sarcoma therapy?
[Response]
Thank you for your comment. The purpose of the study is that low dose RT can be effective enough for skin lesions in CKS. Of course, additional research is needed, but in this study, we analyzed whether there would be a difference in terms of control when the RT dose was high, but we could not confirm the difference according to the dose because the response to the treatment of KS itself was good. It is expected that low dose RT can be useful in that it can control tumor well while reducing side effects and can be used repeatedly in case of recurrence.
- Park, K.; Bae, J.M.; Chung, K.Y.; Yun, S.J.; Seo, S.H.; Ahn, H.H.; Lee, D.Y.; Kim, H.; Sohn, U.; Park, B.C. Incidence and Prevalence of Skin Cancers in South Korea from 2008 to 2016: A Nation-Wide Population Based Study. Ann Dermatol 2022, 34, 105-109, doi:10.5021/ad.2022.34.2.105.
- Park, B.; Ahn, K.H.; Choi, Y.; Kim, J.H.; Seong, H.; Kim, Y.J.; Choi, J.Y.; Song, J.Y.; Lee, E.; Jun, Y.H.; et al. Cancer Incidence Among Adults With HIV in a Population-Based Cohort in Korea. JAMA Netw Open 2022, 5, e2224897, doi:10.1001/jamanetworkopen.2022.24897.
Round 2
Reviewer 2 Report
Comments and Suggestions for Authors
I have no further comments